# Extraction, analysis, and antifungal activity study of algae antibiotic active substances in plateau lakes

Jing Zhao[1], Ju Tang[1‡], Zhandi Wang[1], Shahzad Munir[2], Rong Jiao[1], Jia Li[1], Kaixiang Chao[1], Qionglian Wan[1], Lianchun Wang[1], Changbing Ye[1]*

1 School of Chemistry, Biology and Environment, Yuxi Normal University, Yuxi, Yunnan, China, 2 State Key Laboratory for Conservation and Utilization of Bio-Resources in Yunnan, Yunnan Agricultural University, Kunming, China

‡ co-author.
* yechangbing@yxnu.edu.cn

## Abstract

This study was carried out to assess the inhibitory activity of algae in plateau lakes against plant pathogenic fungi, and further conduct preliminary research and analysis on their antifungal active ingredients, in order to provide a certain basis for the development and utilization of algal secondary metabolite as anti-plant pathogenic fungal agents. Different solvent extraction methods using water, ethyl acetate, ethanol, and methanol were conducted to extract polyphenol metabolites from *Ulothrix*, *Chlorella vulgaris*, and *Microcystis pseudofilamentosa* Crow. The composition of the extracts was analyzed based on the UPLC-MS/MS detection platform, and the antifungal activity was determined. The results showed that the content of polyphenol metabolites extracted from *Ulothrix* using methanol was the highest, followed by ethyl acetate and ethanol. The water extraction method resulted in the highest loss of polyphenolic metabolites. Three species of *Fusarium oxysporum* were used as indicator fungus to determine the antifungal activities of algae extracts. The three types of algae extracts showed good antimicrobial effects on *F. oxysporum*. The polyphenol metabolites extracted from *Ulothrix* using methanol demonstrated the strongest antifungal activity, with up to 20 mm in diameter of inhibition zone. The metabolite with the weakest antifungal activity was *M. pseudofilamentosa* Crow, with an 8-mm diameter of inhibition zone. Based on the UPLC-MS/MS detection, 242 polyphenol metabolites were initially identified in the methanol extracts of the three algae, including 160 phenolic acids, 32 flavonoids, 17 flavonols, 7 dihydroflavones, 2 dihydroflavonols, 2 chalcones, 2 flavanols, 5 flavonoids, 5 lignans, and 10 coumarins. Principal component analysis, fold change analysis, and Kyoto Encyclopedia of Genes and Genomes pathway enrichment analysis were used to conduct differential metabolite screening and related metabolic pathway enrichment. The methanol extract samples of the three algae were mainly classified into two categories. Ten important

**Data availability statement:** All relevant data are within the manuscript and its Supporting Information files.

**Funding:** This study was funded by the National Natural Science Foundation of China (grant number 32060036 and 52360030). The funders had no role in study design, data collection and analysis, decision to publish, or preparation of the manuscript.

**Competing interests:** The authors declare that the research was conducted in the absence of any commercial or financial relationships that could be construed as a potential conflict of interest.

differential metabolites and 15 important metabolic pathways were obtained. In addition, the methanol extracts of *Ulothrix* contained the largest variety of phenolic acid compounds, with a total of 75 phenolic acid compounds detected, which was followed by *C. vulgaris*, with 44 phenolic acid compounds detected, and *M. pseudofilamentosa* Crow, with 37 phenolic acid compounds. Compared with phenolic acid compounds, the quantitative differences of other polyphenols were smaller. Based on these results and those from the antifungal experimental analysis, phenolic acids in algae polyphenol metabolites are the main antifungal active ingredients.

## Introduction

Under suitable hydrological conditions such as temperature and light, lake algae can experience explosive growth. The increase in algae can affect water quality, damaging the aquatic environment. Harmless treatment methods for algae primarily include sanitary landfills, anaerobic fermentation to produce biogas for power generation, biofertilizer production, biofuel generation, incineration for power generation, bioplastic production, algal protein extraction, and biochar production [1–3]. Although these methods of treatment provide resource utilization pathways for lake algae, each has certain limitations. Producing biofertilizers from algae may result in algal toxin residues and nitrogen loss, and the high water content of the algae complicates immediate fermentation, increasing operational difficulty during fermentation for biogas and amino acid production; biochar production requires a high reaction temperature, leading to high energy consumption [4–5]. Algae resource utilization must consider both economic feasibility and operational complexity. Finding a new, economically viable pathway for algae resource utilization is an urgent issue in algae treatment.

In recent years, phenolic acid antifungal compounds have attracted increasing attention from researchers worldwide as efficient and environmentally friendly fungicides. In some economic crops, such as chickpeas, peas, cucumbers, and watermelons [6–8], the use of phenolic acids has proven to be effective in controlling root rot disease. Research on its extraction has long focused on higher plants, such as honeysuckle, ginkgo biloba, and xanthium [9]. However, the cumbersome extraction method and high material costs make it challenging to apply phytogenic phenolic acid fungicides on a large scale in agriculture. Therefore, identifying new sources of antifungal phenolic acid compounds has become particularly important for increasing the use of natural antifungal agents to reduce the prevalence of diseases. many active substances with anti-phytopathogenic activities against fungi and bacteria have been discovered in marine macroalgae, and algae bioactive substances have received increasing attention. Algae antibiotic bioactive substances include polysaccharides, indoles, terpenes, acetylcholines, phenolic acids, and fatty acids [1,10–13]. The research results of Scaglioni on phenolic acid compounds of microalgae and macrospirulina showed that the extraction efficiency of phenolic acid compounds in microalgae was higher, which had the strongest antimicrobial effect on *Fusarium* [14]. Sheikh evaluated the antifungal activities of 15 kinds of algae extracts, among which

Chlorophyta had the best antifungal activity [15]. Asimakis summarized advances in research on metabolites of macroalgae and microalgae (including cyanobacteria), confirming that these compounds are effective antibacterial, antiviral, antifungal, nematicides, insecticides, herbicides, and plant growth promoters [16]. Prisa reported that the algae have demonstrated their potential to play a significant role in crop protection and enhancement [17].

*Fusarium oxysporum* E.F. Sm. & Swingle is a soil-borne pathogenic fungus widely distributed around the globe. It has strong pathogenicity and a complex genetic evolution, which allows it to survive in soil for many years. This fungus has a wide host range and can infect more than 100 kinds of crops of economic value, causing wilt or root rot and significantly affecting the growth, development, yield, and quality of plants [18–20]. As a disease management strategy, biological control has good application prospects because it is environmentally friendly. At present, the biological control of *F. oxysporum* mainly focuses on the use of antagonistic organisms, genetically modified plants, and biological secondary metabolite formulations [21–23].

In this study, three species of algae from plateau lakes in Yunnan Province were collected, and the extraction rates and antifungal activities of polyphenol metabolites using different extraction methods were compared and analyzed. *Ulothrix* are common freshwater species with strong environmental adaptability. *Microcystis pseudofilamentosa* Crow and *Chlorella vulgaris* are two dominant species that accompany blue-green algae during algal blooms in lakes. Using the ultra-performance liquid chromatography (UPLC) and tandem mass spectrometry (MS/MS) detection technique, the main component of the methanol extracts of the three algae were analyzed.

## Materials and methods

### Materials

The algae tested were *Ulothrix*, *Microcystis pseudofilamentosa* Crow, and *Chlorella vulgaris*. The three algae were collected from the Water Treatment Technology and Water Environment Ecological Restoration Engineering Research Center of Yuxi Normal University. *F. oxysporum* E.F. Sm. & Swingle, *F. oxysporum* Schl. f. sp. *fragariae*, and *F. oxysporum* f. sp. *melonis* were sourced from the Key Laboratory of Research on Authentic Medicinal Materials in Central Yunnan, Yuxi Normal University.

### Methods

**Extraction of algae polyphenol metabolites.**

**Improved lime sulfur method**: A total of 5 g of algae was combined with 15 times the amount of purified water and extracted for 2 h. The sample was filtered, and then 10 times the amount of water was added to the filter residue. The sample was extracted for 1.5 h. The filtrate was then combined and concentrated. Afterwards, 20% lime milk was slowly added, the pH was adjusted to 10 value, and the sample was centrifuged. The precipitate was combined with twice the amount of 95% ethanol and ground into a slurry. The pH of the sample was adjusted to 3 by adding 50% sulfuric acid and stirring thoroughly. Afterward, the sample was centrifuged at 5000 rpm, and 40% NaOH was added to the supernatant to adjust the pH to 6. The sample was dried under reduced pressure to obtain the crude extracts. The crude extract was then dissolved in 3 times the amount of water, and the pH was adjusted to 2 with hydrochloric acid. Extraction was done with ethyl acetate 3 times. The extracted liquid was decolorized with activated carbon and filtered. After concentrating the filtrate, an appropriate amount of chloroform was added in batches. Extracts were then obtained after filtration, precipitation, and drying at normal pressure.

**Ethyl acetate extraction method**: A total of 5 g of algae was combined with 10 times the amount of ethyl acetate and extracted twice at 50 °C for 2 h each time. The sample was filtered, and the filtrate was combined and concentrated. The pH of the concentrated solution was adjusted to 10 with NaOH. The sample was extracted with n-butanol three times, combining the extracted liquids and adjusting the pH to 6.5. The sample was then dried and concentrated under reduced pressure to obtain the extracts.

**Ethanol reflux extraction method**: A total of 5 g of algae was combined with 10 times the amount of 50% ethanol. The sample was heated and refluxed at 80 °C twice for 1 h each time. The sample was then filtered, and the filtrate was combined and concentrated under reduced pressure. The pH value of the concentrate was adjusted to 2. Ethyl acetate was added for extraction, and then the extracts were combined and washed twice with an appropriate amount of water. The pH was adjusted to 6.5 with 40% NaOH. The aqueous layer was obtained and concentrated under reduced pressure to a thick paste, which was then dried to obtain the extracts.

**Methanol extraction method**: A total of 5 g of algae was combined with 10 times the amount of methanol and was centrifuged for 60 min at 25 °C at 200 rpm. The extracts were rinsed with hexane and allowed to stand to remove the hexane. Extracts were finally obtained after drying with a rotary evaporator.

## Determination of polyphenol metabolite content

Using chlorogenic acid as a standard, the phenolic acid content in the sample was quantified by UV spectrophotometry. Quantitative analysis was based on the characteristic UV-visible light absorption of phenolic acids. Accurately weighed 1 mg of chlorogenic acid standard was dissolved in a volumetric flask to prepare standard solutions with concentrations of 1 µg/mL, 3 µg/mL, 5 µg/mL, 7 µg/mL, and 9 µg/m. The absorbance (A) of the solutions was measured at the characteristic absorption wavelength of 324 nm for phenolic acids. A standard curve was generated by plotting absorbance (A) against chlorogenic acid concentration (C), from which the linear regression equation was derived.

## Determination of precision

According to the measurement method of Park, five aliquots of a 5.0 µg/mL chlorogenic acid reference solution were used for consecutive absorbance measurements (A), and the RSD value was calculated [24].

## Determination of stability

A chlorogenic acid solution with a concentration of 20.0 µg/mL was prepared, and the (A) value was measured every 15 minutes according to the measurement method of Park [24]. Measurements were performed continuously for 2 hours and the RSD value was calculated.

## Investigation of antibacterial effects

A 0.2 mL fungal suspension from each of the three species of *F. oxysporum* (*F. oxysporum* E.F. Sm. & Swingle, *F. oxysporum* Schl. f. sp. *fragariae*, and *F. oxysporum* f. sp. *melonis*) were injected into a sterile petri dish, to which 15 mL of agar medium was added. The solution was mixed well and cooled. A sterile Oxford cup was gently placed in the center of each fungal plate. Different algae extracts in aqueous solutions with a concentration of 50% were injected into the Oxford cup. The plates were cultivated for 72 h at 26 °C, and the diameter of the inhibition zone corresponding to each Oxford cup was measured.

## Determination of algae polyphenol metabolite components

**Chromatography and mass spectrometry acquisition conditions.** The data acquisition instrument system mainly includes the SHIMADZU Nexera X2 for the UPLC and Applied Biosystems 6500 QTRAP for the MS/MS.

For the liquid phase conditions, the chromatographic column Agilent SB-C18 (1.8 µm, 2.1 mm×100 mm) was used. The mobile phase comprised phase A, which was ultrapure water (with 0.1% formic acid), and phase B, which was acetonitrile (with 0.1% formic acid). The gradient elution was as follows: at 0.00 min, the proportion of phase B was 5%; within 9.00 min, the proportion of phase B linearly increased to 95% and maintained at 95% for 1 min; and at 10.00 min to 11.10 min, the proportion of phase B decreased to 5%, and the column was equilibrated at 5% for 14 min. The mobile phase was run at 0.35 mL/min. The column temperature was 40 °C, and the injection volume was 2 µL.

For the mass spectrometry conditions, the electrospray ionization source temperature was 500°C. The ion spray voltage was 5500 V (positive ion mode) and -4500 V (negative ion mode). Ion source gas I, gas II, and curtain gas were set to 50, 60, and 25 psi, respectively, and the collision-induced ionization parameter was set to high. Instrument tuning and mass calibration were performed using 10 and 100 µmol/L polypropylene glycol solutions under QQQ and LIT modes, respectively. The QQQ scan used MRM mode and the collision gas (nitrogen) was set to medium. Through further optimization of declustering potential (DP) and collision energy (CE), the DP and CE of each MRM ion pair were completed. A specific group of MRM ion pair was monitored based on the metabolites eluted during each period.

**Qualitative and quantitative analysis of metabolites.** Based on the database, qualitative analysis of substances was performed using secondary spectral information, while quantitative analysis of metabolites was carried out using the multiple reaction monitoring (MRM) mode of triple quadrupole mass spectrometry (as shown in the figure below). The characteristic ions of each substance were selected by the triple quadrupole, and their signal intensity (CPS) was measured by the detector. MultiQuant software was then used to perform chromatographic peak integration and calibration. The peak area of each chromatographic peak represented the relative content of the corresponding substance.

**Differential metabolites selected.** For two-group analysis, differential metabolites were determined by absolute Log2FC ($|Log2FC| \geq 1.0$).

## Statistical analyses

All experiments were determined in biological triplicate to ensure reproducibility Experimental results were obtained as the mean value ± SD. Statistical analyses were performed using the SPSS statistical package (SPSS Inc., Chicago, IL, USA). Using Turkey test, the statistical significance was achieved when $p < 0.05$.

## Results and discussion

### Extraction efficiency

Using chlorogenic acid as a standard, the polyphenol content of the extract was determined by spectrophotometry. A linear regression equation was obtained by plotting the concentration of chlorogenic acid (C) on the x-axis and the measured absorbance on the y-axis: $y = 0.0515x + 0.0378$ ($r = 0.9987$). This indicated a strong linear relationship for chlorogenic acid within a concentration range of 1–9 µg/ml. The RSD values for the precision and stability experiments were 1.19% and 0.98%, respectively, indicating high precision and stability of the compound within 2 hours.

Among the four solvent extraction methods, the methanol extraction method had the highest polyphenol metabolite content (Table 1). The polyphenol metabolite contents in the extracts of the three algae (*C. vulgaris*, *Ulothrix*, and *M. pseudofilamentosa* Crow) were 3.353, 5.091, and 4.856 µg/mL, respectively. The polyphenol content of the extracts obtained through the ethanol reflux extraction method was the lowest. Comparative analysis of the three algae showed that the content of polyphenol metabolites in *Ulothrix* was higher than that of the other algae. In addition, the extraction agent of the improved lime sulfur method was water, and that of the other three methods

Table 1. Investigation on extraction methods of polyphenol metabolites.

| Polyphenol Content of three algae(µg/mL) | Extraction of algae polyphenol metabolites | | | |
|---|---|---|---|---|
| | Improved lime sulfur method | Ethyl acetate extraction method | Ethanol reflux extraction method | Methanol extraction method |
| **C. *vulgaris*** | 0.269±0.83 | 1.035±0.68 | 3.068±0.76 | 3.353±0.26 |
| ***Ulothrix*** | 0.956±1.25 | 2.955±1.35 | 3.082±0.52 | 5.091±0.12 |
| **M. *pseudofilamentosa* Crow** | 0.685±0.63 | 3.569±0.26 | 2.063±1.03 | 4.856±0.48 |

was organic solvent. Based on the polyphenol content, the polyphenol material loss of the water extraction method may have been high or the extraction rate may be poor. In preliminary literature reports, the agents used for the extraction of antifungal active substances from algae include water, methanol, ethanol, ethyl acetate, chloroform, acetone, and hexane [25–26]. Among these, methanol, acetone, and ethyl acetate exhibited the highest extraction efficiencies. Khadija used four solvents, including methanol, to extract and prepare marine red algae extracts, among which the methanol extract had the strongest antifungal and insect resistance activity [27]. Scaglioni prepared 15 marine macroalgae using six organic solvents, including acetone and methanol [11]. The acetone extract showed the strongest antifungal activity, with an antifungal activity of 19.3%. In this study, UPLC-MS/MS was used to analyze the components of three algal methanol extracts and 242 polyphenolic metabolites were preliminarily identified, of which 160 were phenolic acids (66%). It was speculated that phenolic acids in algae are the primary active antifungal ingredients.

## Antifungal activity

The plate confrontation test was adopted to analyze the antifungal activities of the algae polyphenol metabolites. Three species of *F. oxysporum* (*F. oxysporum* E.F. Sm. & Swingle, *F. oxysporum* Schl. f. sp. fragariae, and *F. oxysporum* f. sp. *melonis*) were used as indicator fungi to determine the antifungal activities of the polyphenol metabolite extracts of the algae. The results showed that the extracts of *Ulothrix*, *C. vulgaris*, and *M. pseudofilamentosa* Crow had strong antimicrobial effects on *F. oxysporum* E.F. Sm. & Swingle. The polyphenol metabolites of *Ulothrix* extracted using methanol had the widest antifungal activity range, with an antifungal diameter of as wide as 20 mm. The *M. pseudofilamentosa* Crow extracts had the weakest antifungal activity, with an antimicrobial zone diameter of 6.5 mm to 9 mm against the three species of *F. oxysporum* (S1 Fig).

## Metabolite identification

The UPLC-MS/MS was used to detect and analyze the components of the methanol extracts of the three algae. The total ion chromatogram of the chemical composition characterization of the mixed sample of methanol extracts of the three algae analyzed by mass spectrometry is shown in S2 Fig. Based on the metabolic database, qualitative and quantitative analysis of metabolites in the sample was conducted using mass spectrometry. The S2 Fig shows the substances that were detected in the samples. Each mass spectrum peak of different colors represents a detected metabolite. The experiment preliminarily identified 242 polyphenol metabolites, including 160 phenolic acids, 32 flavonoids, 17 flavonols, 7 dihydroflavones, 2 dihydroflavonols, 2 chalcones, 2 flavanols, 5 other flavonoids, 5 lignans, and 10 coumarins. Among these metabolites, 10 important components were analyzed (Table 2).

## Screening of differential polyphenol metabolites

Variable importance in projection (≥ 1) and fold change analysis (≥ 2 or ≤ 0.5) were used to screen the differential polyphenol metabolites. A total of 206 differential extracts were obtained from the polyphenol metabolites of the three algae. Between *Ulothrix* and *M. pseudofilamentosa* Crow, 58 differential metabolites were up-regulated and 109 differential metabolites were down-regulated. Between *M. pseudofilamentosa* Crow and *C. vulgaris*, 42 differential metabolites were up-regulated and 135 differential metabolites were down-regulated. Between *M. pseudofilamentosa* Crow and *C. vulgaris*, 28 differential metabolites were up-regulated and 100 differential metabolites were down-regulated (S3 Fig). The top 10 important differential metabolites are shown in Table 3. Based on the Kyoto Encyclopedia of Genes and Genomes pathway enrichment analysis of differential metabolites, algae polyphenol extracts play a significant role in multiple important pathways such as flavone and flavonol biosynthesis, phenylalanine metabolism, tyrosine metabolism, flavonoid biosynthesis, and phenylpropanoid biosynthesis. The top 15 metabolic pathways are shown in S4 Fig.

**Table 2. Ten important differential metabolites in polyphenol metabolites of three algae.**

| Compounds | Formula | Class | Type |
|---|---|---|---|
| Digallic Acid | $C_{14}H_{10}O_9$ | Phenolic acids | down |
| 1,2,6-Tri-O-galloyl-β-D-glucose | $C_{27}H_{24}O_{18}$ | Phenolic acids | down |
| 2-Naphthol | $C_{10}H_8O$ | Phenolic acids | up |
| 3-O-Galloyl-D-glucose | $C_{13}H_{16}O_{10}$ | Phenolic acids | down |
| Hydrocinnamic acid | $C_9H_{10}O_2$ | Phenolic acids | up |
| 4-O-Glucosyl-4-hydroxybenzoic acid | $C_{13}H_{16}O_8$ | Phenolic acids | down |
| Protocatechuic acid-4-O-glucoside | $C_{13}H_{16}O_9$ | Phenolic acids | down |
| 4-Hydroxy-3,5-dimethoxybenzyl alcohol | $C_9H_{12}O_4$ | Phenolic acids | down |
| Methyl gallate | $C_8H_8O_5$ | Phenolic acids | down |
| 5-O-Galloyl-D-hamamelose | $C_{13}H_{16}O_{10}$ | Phenolic acids | down |

Note:Formula: Molecular formula of a substance; Compounds: Name of the substance; Class: Primary classification of substances; Type: Metabolite up-down type.

**Table 3. Quantity of polyphenol metabolites of three algae.**

| Substance Category | *Ulothrix* | *M.pseudofilamentosa Crow* | *C. vulgaris* |
|---|---|---|---|
| Phenolic acids | 75 | 37 | 44 |
| Flavones | 13 | 8 | 10 |
| Flavonols | 9 | 5 | 8 |
| Flavanones | 2 | 3 | 3 |
| Lignans | 2 | 3 | 1 |
| Other Flavonoids | 2 | 2 | 1 |
| Coumarins | 0 | 4 | 0 |
| Flavanols | 2 | 0 | 0 |
| Chalcones | 1 | 0 | 1 |
| Flavanonols | 0 | 1 | 0 |

Algae, and microalgae in particular, have high biomass, a short growth cycle, high-efficiency photosynthesis, and a high ester content. Compared to crops, their yield per unit area can be multiple times higher. For example, the biomass productivity of different Chlorella strains ranged 0.18–0.34 g $L^{-1}$ $day^{-1}$ [28]. Kim reported that the specific growth rate and biomass productivity of *Chlorella sp*. under a $CO_2$ concentration 0.04% and 100 µmol $m^{-2}$ $s^{-2}$ were 0.50 µ $day^{-1}$ and 0.24 g $L^{-1}$ $day^{-1}$ [29]. The use of algae from plateau lakes to prepare phenolic acid bactericides involves a simple and energy-efficient process. This addresses the problem of large-scale resource utilization of plateau lake algae in addition to solving the issue of the high cost of plant-derived antimicrobial phenolic acid preparations, which has hindered their large-scale application. In the industrial large-scale application of this process, controlling the extraction temperature is crucial to efficiently obtain the antibacterial active components. Algal secondary metabolites are rich in variety, including proteins, polysaccharides, and lipids in addition to phenolic acids. In the next step of our studies, we will further examine the antibacterial activity of other secondary metabolites to enhance the development and utilization of algal antimicrobial agents.

## Conclusions

Phytogenic phenolic acid fungicides are difficult to apply on a large scale because of the complexity of the extraction methods and high material costs. Algae, especially microalgae, possess several advantages such as large biomass, short growth cycles, high photosynthetic reaction efficiency, and abundance of esters. Compared with crops, the yield per unit

area can be 3 times higher. In this study, three types of lake algae all had a good inhibitory effect on the growth of *F. oxysporum*. The extracted polyphenolic metabolites contained abundant phenolic acids, which could be used as a source of phenolic acid fungicides for plant diseases. Using algae in the preparation of antifungal phenolic acid biologics for plateau lakes can (1) being a promising option and difficulties in large-scale application of phytogenic antifungal phenolic acid preparations and (2) solve the problem of large-scale resource utilization of algae in plateau lakes.

## Supporting information

**S1 Fig. Antifungal activity of the different extracts from the three algae.** Note: The vertical axis is inhibition zone(mm), and the horizontal axis is extraction menthods; I: Improved lime sulfur method; II: Ethyl acetate extraction method; III: Ethanol reflux extraction method IV: Methanol extraction method. Different colors represent different algae. Data are expressed as means ± SD of four separate experiments. *P < 0.05, **P < 0.01, ***P < 0.001, when compared to the control.
(TIF)

**S2 Fig. Total ion chromatogram of mass spectrometry analysis of the mixed sample of methanol extracts of three algae.** Note: A. Mixed sample of methanol extracts of the algae (positive ion mode), and B. Mixed sample of methanol extracts of the algae (negative ion mode).
(TIF)

**S3 Fig. Differential gene statistics of three algae.** Number of differentially metabolities identified in three comparision groups.
(TIF)

**S4 Fig. KEGG enrichment analysis diagram of differential metabolites.** Note: The abscissa represents the rich factor corresponding to each pathway. The ordinate represents the pathway name (sorted according to P-value). The color of the point represents the P-value size, with greater color intensity indicating a more significant enrichment. The size of the dots represents the number of enriched differential metabolites, the larger the dot, the greater the number of metabolites(P-value/Count).
(TIF)

**S1 File. The Minimal date of Fig 1.**
(XLSX)

## Acknowledgments

The help of Sangon Biotech in construction of the experimental platform is gratefully acknowledged.

## Author contributions

**Data curation:** Jia Li, Kaixiang Chao.

**Formal analysis:** Rong Jiao, Zhandi Wang, Ju Tang.

**Funding acquisition:** Jing Zhao.

**Investigation:** Qionglian Wan.

**Project administration:** Lianchun Wang.

**Writing – original draft:** Jing Zhao, Changbing Ye.

**Writing – review & editing:** Shahzad Munir.

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
