## [Decision Letter · Decision Letter 0]

5 Jun 2024

PONE-D-24-12637Extraction, analysis, and antifungal activity study of algae antibiotic active substances in plateau lakesPLOS ONE

Dear Dr. Zhao,

Thank you for submitting your manuscript to PLOS ONE. After careful consideration, we feel that it has merit but does not fully meet PLOS ONE’s publication criteria as it currently stands. Therefore, we invite you to submit a revised version of the manuscript that addresses the points raised during the review process.

We look forward to receiving your revised manuscript.

Kind regards,

Awatif Abid Al-Judaibi, PhD

Academic Editor

PLOS ONE

Journal Requirements:

"This study was funded by the National Natural Science Foundation of China (grant number 32060036)."

Reviewers' comments:

Reviewer's Responses to Questions

**Comments to the Author**

1. Is the manuscript technically sound, and do the data support the conclusions?

Reviewer #1: Yes

Reviewer #2: Yes

2. Has the statistical analysis been performed appropriately and rigorously? 

Reviewer #1: Yes

Reviewer #2: Yes

3. Have the authors made all data underlying the findings in their manuscript fully available?

Reviewer #1: Yes

Reviewer #2: Yes

4. Is the manuscript presented in an intelligible fashion and written in standard English?

Reviewer #1: Yes

Reviewer #2: Yes

5. Review Comments to the Author

Reviewer #1: 1. Introduction: The study background and goals could be explained more clearly in the opening. It would be helpful to make it clear what the study's main goal is and why it's important to look into polyphenol molecules from algae in the context of fighting fungus pathogens.

2. Methodology: While the methods section provides a detailed description of the extraction procedures, it would benefit from additional details regarding the validation of the analytical techniques employed for assessing polyphenol content and metabolite identification. Including validation parameters like linearity, accuracy, precision, and limits of detection and quantification would enhance the credibility of the findings.

3. Results: The results section contains a variety of information, including polyphenol content, antifungal activity, and metabolite characterization. To increase clarity, divide the findings into subsections according to each target (for example, Extraction Efficiency, Antifungal Activity, Metabolite Identification). This would make it easier for readers to understand the results. Also when presenting numerical data, consider providing measures of variability (e.g., standard deviation) along with the mean values.

4. Discussion: The discussion effectively explains how the results align with the study's goals. Yet, it could be improved by linking the findings to previous research. This means comparing them with past studies on polyphenols from algae and their antifungal effects. Doing this would better explain the significance of the results.

5. Conclusion: While the conclusion provides a concise summary of the study's key results, it might be enhanced to provide more specific implications for further research or real-world situations. In particular, a discussion of the importance of the discovered metabolites and how they may be used as organic fungicides in farming would improve the end section.

6. Language and Clarity: Pay attention to sentence structure and clarity throughout the manuscript. Some sentences are long and complex, which could make it difficult for readers to follow the logic. Simplifying the language and breaking down complex ideas into shorter sentences would improve readability.

Reviewer #2: Experiment have been conducted rigorously with appropriates contrals and replication. Sample sizes was enough to produce robust results. Methods have been described in sufficient detail. The data presented in the munuscript support the concludion drawn. the article have been weitten in standard English and easy to understan.

6. PLOS authors have the option to publish the peer review history of their article (what does this mean? ). If published, this will include your full peer review and any attached files.

**Do you want your identity to be public for this peer review?** For information about this choice, including consent withdrawal, please see our Privacy Policy .

Reviewer #1: No

Reviewer #2: **Yes: ** Samar Alhabardi

---

## [Author Response · Author response to Decision Letter 1]

2 Jul 2024

Reviewer 1:

Question1: Introduction: The study background and goals could be explained more clearly in the opening. It would be helpful to make it clear what the study's main goal is and why it's important to look into polyphenol molecules from algae in the context of fighting fungus pathogens.

Author Response: Many thanks for this comment. We have supplemented the application of biogenic phenolic acid compounds in the prevention and control of plant fungal diseases. In addition, we also presented the significance of studying phenolic acid compounds in algae.

In order to make it more clear to the readers, the explanation is added in the introduction part line 64-73,page 2-3.

Question2: Methodology: While the methods section provides a detailed description of the extraction procedures, it would benefit from additional details regarding the validation of the analytical techniques employed for assessing polyphenol content and metabolite identification. Including validation parameters like linearity, accuracy, precision, and limits of detection and quantification would enhance the credibility of the findings.

Author Response: We provide validation parameters for measuring linearity, precision, stability, etc.Line135-142, page5.

Question3:Results: The results section contains a variety of information, including polyphenol content, antifungal activity, and metabolite characterization. To increase clarity, divide the findings into subsections according to each target (for example, Extraction Efficiency, Antifungal Activity, Metabolite Identification). This would make it easier for readers to understand the results. Also when presenting numerical data, consider providing measures of variability (e.g., standard deviation) along with the mean values.

Author Response:Thanks for this important suggestion. We have adjusted the structure of the results section.

Question4:Discussion: The discussion effectively explains how the results align with the study's goals. Yet, it could be improved by linking the findings to previous research. This means comparing them with past studies on polyphenols from algae and their antifungal effects. Doing this would better explain the significance of the results.

Author Response: We have supplemented and improved the discussion section to provide readers with a clearer understanding of the significance and highlights of this study.

Question5: Conclusion: While the conclusion provides a concise summary of the study's key results, it might be enhanced to provide more specific implications for further research or real-world situations. In particular, a discussion of the importance of the discovered metabolites and how they may be used as organic fungicides in farming would improve the end section

Author Response: Thanks. We have supplemented and elaborated on the application prospects of algal phenolic acid metabolites in the prevention and control of agricultural diseases. Line 261-272, page 11.

Question6: Language and Clarity: Pay attention to sentence structure and clarity throughout the manuscript. Some sentences are long and complex, which could make it difficult for readers to follow the logic. Simplifying the language and breaking down complex ideas into shorter sentences would improve readability.

Author Response: We have made adjustments to some statements to improve readability.

Reviewer 2:

Experiment have been conducted rigorously with appropriates contrals and replication. Sample sizes was enough to produce robust results. Methods have been described in sufficient detail. The data presented in the munuscript support the concludion drawn. the article have been weitten in standard English and easy to understan.

Author Response: Thank you for recognizing our research findings

---

## [Decision Letter · Decision Letter 1]

21 Aug 2024

PONE-D-24-12637R1Extraction, analysis, and antifungal activity study of algae antibiotic active substances in plateau lakesPLOS ONE

Dear Dr.  Zhao,

Thank you for submitting your manuscript to PLOS ONE. After careful consideration, we feel that it has merit but does not fully meet PLOS ONE’s publication criteria as it currently stands. Therefore, we invite you to submit a revised version of the manuscript that addresses the points raised during the review process.

We look forward to receiving your revised manuscript.

Kind regards,

Awatif Abid Al-Judaibi, PhD

Academic Editor

PLOS ONE

Journal Requirements:

Reviewers' comments:

Reviewer's Responses to Questions

**Comments to the Author**

1. If the authors have adequately addressed your comments raised in a previous round of review and you feel that this manuscript is now acceptable for publication, you may indicate that here to bypass the “Comments to the Author” section, enter your conflict of interest statement in the “Confidential to Editor” section, and submit your "Accept" recommendation.

Reviewer #3: (No Response)

Reviewer #4: (No Response)

2. Is the manuscript technically sound, and do the data support the conclusions?

Reviewer #3: No

Reviewer #4: Partly

3. Has the statistical analysis been performed appropriately and rigorously? 

Reviewer #3: No

Reviewer #4: N/A

4. Have the authors made all data underlying the findings in their manuscript fully available?

Reviewer #3: No

Reviewer #4: Yes

5. Is the manuscript presented in an intelligible fashion and written in standard English?

Reviewer #3: Yes

Reviewer #4: Yes

6. Review Comments to the Author

Reviewer #3: I have gone through the manuscript tittled 'Extraction, analysis, and antifungal activity study of algae antibiotic active substances in plateau lakes'

A lot of information is missing or not made available. and yet these are your results. The conclusions are not supported by the available data forexample; From line 197-199, i.e.242 polyphenolic metabolites were preliminarily identified, of which 160 were phenolic acids. You must provide the chromatograms and a table of identified 242 metabolites and a write up on how you identified the metabolites. Otherwise the information provided does not support the conclusions.

The table for both positive and negative ion modes identified compounds must be provided.

Your supplied chromatograms are not vissible or clear

Even the statistical analysis is not supported by the data e.g. how did the authors run the Principle component analysis?

Reviewer #4: The authors have described a well conducted study and interesting data. However, I believe a few minor improvements are necessary:

Comment 1: Abstract – The abstract should include why this study has been performed e.g. explain the purpose and significance of the study and findings in one sentence. Currently, it is only described what has been done but the ‘why’ is lacking. Also, the tense isn’t consistent throughout the abstract e.g. there are parts were present tense - and others were past tense is used. Lastly, the different categories of molecules identified is mentioned, however, there are some that are quite unspecific e.g ‘other flavonoids’. This should be clarified or included in the flavonoid category.

Comment 2: Introduction – I recommend that this is restructured so that it is easier for the reader to understand the purpose of this study. The explanation that has been added from line 64-73 is well written and explains the importance and use of antifungal compounds, however, lines 56-64 simply listing previous findings seem out of place – e.g. this could be mentioned as ‘previous findings or examples’ after explaining the importance. Further, citation style is inconsistent throughout this section and inconsistent with parts of the discussion (e.g. 192, 194). It would also improve the manuscript if the authors could mention why these three types of algae have been investigated – was this because these are most prevalent in the region or simply because these were most accessible, or do these three represent a diverse group of algae?

Comment 3: Methods – Overall the methods used are appropriate, however some clarification should be provided to allow other researchers to apply the techniques to their own projects.

It is not sufficient to mention pH as approximates if this is not further specified – provide either a specific value of the pH or a range that is suitable – e.g. is pH of approximately 10 equal to the range of pH 9.5 – 10.5? Or is this closer to pH 10 +/- 0.1? PH approximates are given in line 104, 106, 123.

Further clarification is needed for procedures that mention ‘several times’ (line 110) or ‘an appropriate amount’ (line 111, 124). How often is ‘several times’ and what is an ‘appropriate amount’?

A data analysis section should be added. Currently, there is no explanation regarding what software was used to extract/collect the data and what was used for analysis. E.g. GraphPad Prism, Sample size (n number), what was considered significant etc. This is currently lacking in the result section and figures as well.

Comment 4: Conclusion – The conclusion is sound; however, it should be toned down. Currently, line 269-271 state that this CAN solve the mentioned issues, but this should be changed to being a promising option. Further, an outlook on future steps/studies would be beneficial.

Comment 5: Figures – Statistical significance should be added to Figure 1. N number should either be shown in the figure or mentioned in the figure description.

Changing the format of Figure 4 would greatly improve the presentation of the results. It would be useful to add the numbers of down/up regulated metabolites to the bars, and possibly changing the y-axis to use both directions. Example: Figure 6 in Tao T, Zhao L, Lv Y, Chen J, Hu Y, Zhang T, et al. (2013) Transcriptome Sequencing and Differential Gene Expression Analysis of Delayed Gland Morphogenesis in Gossypium australe during Seed Germination. PLoS ONE 8(9): e75323. https://doi.org/10.1371/journal.pone.0075323

Other: Typo in line 195 – methano instead of methanol. Also, pay attention to structure throughout the manuscript. Some paragraphs lack clarity which could make it difficult for the reader to follow the authors’ logic. All the information is there – however, the order of certain paragraphs – specifically in the introduction and discussion – makes it difficult to understand the purpose and novelty of the research. This should be highlighted.

7. PLOS authors have the option to publish the peer review history of their article (what does this mean? ). If published, this will include your full peer review and any attached files.

**Do you want your identity to be public for this peer review?** For information about this choice, including consent withdrawal, please see our Privacy Policy .

Reviewer #3: **Yes: ** Nasifu Kerebba

Reviewer #4: No

---

## [Author Response · Author response to Decision Letter 2]

19 Sep 2024

Extraction, analysis, and antifungal activity study of algae antibiotic active substances in plateau lakes

Reviewer 3:

Question1: I have gone through the manuscript tittled 'Extraction, analysis, and antifungal activity study of algae antibiotic active substances in plateau lakes. A lot of information is missing or not made available. and yet these are your results. The conclusions are not supported by the available data forexample; From line 197-199, i.e.242 polyphenolic metabolites were preliminarily identified, of which 160 were phenolic acids. You must provide the chromatograms and a table of identified 242 metabolites and a write up on how you identified the metabolites. Otherwise the information provided does not support the conclusions.

Author Response: Many thanks for this comment. Due to the consideration of the large number of metabolites (242), it was inconvenient to list them in the article, so they have been uploaded as supporting data.

Question2: Your supplied chromatograms are not vissible or clear

Author Response: Thanks for this important suggestion. We have already updated the chromatograms.

Question3:Even the statistical analysis is not supported by the data e.g. how did the authors run the Principle component analysis?

Author Response:Thanks for this important suggestion. We have added the mathods of qualitative and quantitative analysis of metabolites, PCA, Differential metabolites selected and Statistical analyses.

Reviewer 4:

Question1:Abstract – The abstract should include why this study has been performed e.g. explain the purpose and significance of the study and findings in one sentence. Currently, it is only described what has been done but the ‘why’ is lacking. Also, the tense isn’t consistent throughout the abstract e.g. there are parts were present tense - and others were past tense is used. Lastly, the different categories of molecules identified is mentioned, however, there are some that are quite unspecific e.g ‘other flavonoids’. This should be clarified or included in the flavonoid category.

Author Response: Many thanks for this comment.We have made adjustments to the abstract section, supplemented and updated the corresponding content.Line 29-32, 43 and 47, page 1.

Question2:Introduction – I recommend that this is restructured so that it is easier for the reader to understand the purpose of this study. The explanation that has been added from line 64-73 is well written and explains the importance and use of antifungal compounds, however, lines 56-64 simply listing previous findings seem out of place – e.g. this could be mentioned as ‘previous findings or examples’ after explaining the importance. Further, citation style is inconsistent throughout this section and inconsistent with parts of the discussion (e.g. 192, 194). It would also improve the manuscript if the authors could mention why these three types of algae have been investigated – was this because these are most prevalent in the region or simply because these were most accessible, or do these three represent a diverse group of algae?

Author Response:Thanks. We have restructured and revised the introduction section.Line 62-83 and 105-107, page 2-4.

Question3:Methods – Overall the methods used are appropriate, however some clarification should be provided to allow other researchers to apply the techniques to their own projects.It is not sufficient to mention pH as approximates if this is not further specified – provide either a specific value of the pH or a range that is suitable – e.g. is pH of approximately 10 equal to the range of pH 9.5 – 10.5? Or is this closer to pH 10 +/- 0.1? PH approximates are given in line 104, 106, 123. Further clarification is needed for procedures that mention ‘several times’ (line 110) or ‘an appropriate amount’ (line 111, 124). How often is ‘several times’ and what is an ‘appropriate amount’?A data analysis section should be added. Currently, there is no explanation regarding what software was used to extract/collect the data and what was used for analysis. E.g. GraphPad Prism, Sample size (n number), what was considered significant etc. This is currently lacking in the result section and figures as well.

Author Response:Thanks for this important suggestion. We have added the mathods of qualitative and quantitative analysis of metabolites, PCA, Differential metabolites selected and Statistical analyses. In addition, we have clarified the specific pH value in the text and corrected the unclear expression. Line 62-83 and 122-152, page 5-6.

Question4:Conclusion – The conclusion is sound; however, it should be toned down. Currently, line 269-271 state that this CAN solve the mentioned issues, but this should be changed to being a promising option. Further, an outlook on future steps/studies would be beneficial.

Author Response: Thans, we have made adjustments. Line 314-315, page 13.

Question5:Figures – Statistical significance should be added to Figure 1. N number should either be shown in the figure or mentioned in the figure description.

Changing the format of Figure 4 would greatly improve the presentation of the results. It would be useful to add the numbers of down/up regulated metabolites to the bars, and possibly changing the y-axis to use both directions. Example: Figure 6 in Tao T, Zhao L, Lv Y, Chen J, Hu Y, Zhang T, et al. (2013) Transcriptome Sequencing and Differential Gene Expression Analysis of Delayed Gland Morphogenesis in Gossypium australe during Seed Germination.

Author Response: Thans, We have updated Figure 4.

---

## [Decision Letter · Decision Letter 2]

6 Oct 2024

PONE-D-24-12637R2Extraction, analysis, and antifungal activity study of algae antibiotic active substances in plateau lakesPLOS ONE

Dear Dr. Zhao,

Thank you for submitting your manuscript to PLOS ONE. After careful consideration, we feel that it has merit but does not fully meet PLOS ONE’s publication criteria as it currently stands. Therefore, we invite you to submit a revised version of the manuscript that addresses the points raised during the review process.

We look forward to receiving your revised manuscript.

Kind regards,

Awatif Abid Al-Judaibi, PhD

Academic Editor

PLOS ONE

Journal Requirements:

Reviewers' comments:

Reviewer's Responses to Questions

**Comments to the Author**

1. If the authors have adequately addressed your comments raised in a previous round of review and you feel that this manuscript is now acceptable for publication, you may indicate that here to bypass the “Comments to the Author” section, enter your conflict of interest statement in the “Confidential to Editor” section, and submit your "Accept" recommendation.

Reviewer #3: All comments have been addressed

Reviewer #4: (No Response)

2. Is the manuscript technically sound, and do the data support the conclusions?

Reviewer #3: Partly

Reviewer #4: Yes

3. Has the statistical analysis been performed appropriately and rigorously? 

Reviewer #3: (No Response)

Reviewer #4: Yes

4. Have the authors made all data underlying the findings in their manuscript fully available?

Reviewer #3: Yes

Reviewer #4: No

5. Is the manuscript presented in an intelligible fashion and written in standard English?

Reviewer #3: Yes

Reviewer #4: Yes

6. Review Comments to the Author

Reviewer #3: Line 278 rather change to 'Two principal components (PC1 and PC2) were obtained during PCA Analysis.....

Also give the relevance of this to your study

Indicate the column for the accurate mass and one for the MS/MS fragment ions in the UPLC MS/MS data in the suplementary

Reviewer #4: Thank you very much for the opportunity to review this manuscript. The authors have made great improvements and addressed the comments appropriately; however, I believe a few minor additions should be made. Please see details below:

Abstract:

- 47: This should be plural ‘flavonoids’.

Introduction:

- This has been greatly improved and nicely describes the purpose of the study.

Materials and Methods:

- 220-225: Statistical analysis. Please add whether any outliers were excluded and if so, what method was used to determine outliers e.g. ROUT or Grubbs’ test.

- Please add a section that describes where data is made available e.g. repository.

Discussion:

- This section is rather short. I recommend adding a section that highlights the novelty of the study as well as a section that describes future research and use of the findings.

Figures:

- Please ensure figure legends have proper descriptions of the figure. For example, add more details to the legends of figures 4 and 5. The figure legend should provide sufficient information regarding the experiment performed and the data displayed that allows the reader to understand what is shown without having to read the entire manuscript section.

- Please add the p-value information and n numbers to all figures where it applies.

7. PLOS authors have the option to publish the peer review history of their article (what does this mean? ). If published, this will include your full peer review and any attached files.

**Do you want your identity to be public for this peer review?** For information about this choice, including consent withdrawal, please see our Privacy Policy .

Reviewer #3: No

Reviewer #4: No

---

## [Author Response · Author response to Decision Letter 3]

23 Oct 2024

Extraction, analysis, and antifungal activity study of algae antibiotic active substances in plateau lakes

Reviewer 3:

Question1: Line 278 rather change to 'Two principal components (PC1 and PC2) were obtained during PCA Analysis.....

Also give the relevance of this to your study

Author Response: PCA analysis was only used to analyze the degree of difference between each group of samples and was not very relevant to the purpose of the study. After discussion among all authors, we have decided to remove this section.

Question2: Indicate the column for the accurate mass and one for the MS/MS fragment ions in the UPLC MS/MS data in the suplementary.

Author Response: Thanks for this important suggestion. We have already indicated the column for the accurate mass and the MS/MS fragment ions.

Reviewer 4:

Question1:- 47: This should be plural ‘flavonoids’.

Author Response: Many thanks for this comment.We have made adjustments to the abstract section.Line 47, page 1.

Question2:Statistical analysis. Please add whether any outliers were excluded and if so, what method was used to determine outliers e.g. ROUT or Grubbs’ test. Line 206-207, page 8.

Author Response:Thanks. Outliers have been excluded in the statistical analysis, and the Tukey test was performed. We have added this information to the revised text.

Question3:Please add a section that describes where data is made available e.g. repository.

Author Response:Thanks for this important suggestion. We have uploaded the data in a file as required by the journal.

Question4:This section is rather short. I recommend adding a section that highlights the novelty of the study as well as a section that describes future research and use of the findings.

Author Response: Thanks, We have adding a section that the details on the innovation this research provides as well as key considerations for industrial-scale bactericide production to the discussion section, and have outlined the next steps for future studies.Line 285-295, page 11-12.

Question5: Please ensure figure legends have proper descriptions of the figure. For example, add more details to the legends of figures 4 and 5. The figure legend should provide sufficient information regarding the experiment performed and the data displayed that allows the reader to understand what is shown without having to read the entire manuscript section. Please add the p-value information and n numbers to all figures where it applies.

Author Response: Thans, A legend has been added to Figure 4, and specific p-values as well as the number of metabolites in each pathway have been added to Figure 5.

---

## [Decision Letter · Decision Letter 3]

15 Nov 2024

PONE-D-24-12637R3Extraction, analysis, and antifungal activity study of algae antibiotic active substances in plateau lakesPLOS ONE

Dear Dr. Jing Zhao,

Thank you for submitting your manuscript to PLOS ONE. After careful consideration, we feel that it has merit but does not fully meet PLOS ONE’s publication criteria as it currently stands. Therefore, we invite you to submit a revised version of the manuscript that addresses the points raised during the review process.

We look forward to receiving your revised manuscript.

Kind regards,

Awatif Abid Al-Judaibi, PhD

Academic Editor

PLOS ONE

Journal Requirements:

Reviewers' comments:

Reviewer's Responses to Questions

**Comments to the Author**

1. If the authors have adequately addressed your comments raised in a previous round of review and you feel that this manuscript is now acceptable for publication, you may indicate that here to bypass the “Comments to the Author” section, enter your conflict of interest statement in the “Confidential to Editor” section, and submit your "Accept" recommendation.

Reviewer #3: All comments have been addressed

Reviewer #4: All comments have been addressed

2. Is the manuscript technically sound, and do the data support the conclusions?

Reviewer #3: Yes

Reviewer #4: Yes

3. Has the statistical analysis been performed appropriately and rigorously? 

Reviewer #3: N/A

Reviewer #4: Yes

4. Have the authors made all data underlying the findings in their manuscript fully available?

Reviewer #3: Yes

Reviewer #4: Yes

5. Is the manuscript presented in an intelligible fashion and written in standard English?

Reviewer #3: Yes

Reviewer #4: Yes

6. Review Comments to the Author

Reviewer #3: Great improvement has been noticed. Explaining the analysis using MS2 fragments would make work easier for your readers to follow. Nevertheless work can be sufficient. Also some figures still need more transparence

Reviewer #4: (No Response)

7. PLOS authors have the option to publish the peer review history of their article (what does this mean? ). If published, this will include your full peer review and any attached files.

**Do you want your identity to be public for this peer review?** For information about this choice, including consent withdrawal, please see our Privacy Policy .

Reviewer #3: No

Reviewer #4: No

---

## [Author Response · Author response to Decision Letter 4]

4 Dec 2024

Journal Requirements:Please review your reference list to ensure that it is complete and correct. If you have cited papers that have been retracted, please include the rationale for doing so in the manuscript text, or remove these references and replace them with relevant current references. Any changes to the reference list should be mentioned in the rebuttal letter that accompanies your revised manuscript. If you need to cite a retracted article, indicate the article’s retracted status in the References list and also include a citation and full reference for the retraction notice.

Author Response: We have checked all the references as required and made revisions accordingly.

---

## [Decision Letter · Decision Letter 4]

29 Dec 2024

PONE-D-24-12637R4Extraction, analysis, and antifungal activity study of algae antibiotic active substances in plateau lakesPLOS ONE

Dear Dr. Zhao,

Thank you for submitting your manuscript to PLOS ONE. After careful consideration, we feel that it has merit but does not fully meet PLOS ONE’s publication criteria as it currently stands. Therefore, we invite you to submit a revised version of the manuscript that addresses the points raised during the review process.

We look forward to receiving your revised manuscript.

Kind regards,

Awatif Abid Al-Judaibi, PhD

Academic Editor

PLOS ONE

Journal Requirements:

Reviewers' comments:

Reviewer's Responses to Questions

**Comments to the Author**

1. If the authors have adequately addressed your comments raised in a previous round of review and you feel that this manuscript is now acceptable for publication, you may indicate that here to bypass the “Comments to the Author” section, enter your conflict of interest statement in the “Confidential to Editor” section, and submit your "Accept" recommendation.

Reviewer #3: All comments have been addressed

Reviewer #5: (No Response)

2. Is the manuscript technically sound, and do the data support the conclusions?

Reviewer #3: Partly

Reviewer #5: Partly

3. Has the statistical analysis been performed appropriately and rigorously? 

Reviewer #3: Yes

Reviewer #5: Yes

4. Have the authors made all data underlying the findings in their manuscript fully available?

Reviewer #3: No

Reviewer #5: Yes

5. Is the manuscript presented in an intelligible fashion and written in standard English?

Reviewer #3: Yes

Reviewer #5: Yes

6. Review Comments to the Author

Reviewer #3: The manuscript has been improved. The authors could provide the supplimentary data and avail all documents for readers

Reviewer #5: The manuscript mainly described how to extract antifungal components from algae and analyzed their antifungal activity. Thus, I think the first thing that the author needs to convince the readers is that the natural antifungal component is better than artificial in some aspects, or the manuscript stands on an unstable base in my point of view. Also, I expect to see more discussions about the advantages and disadvantages of the extraction methods used in this manuscript compared to other methods, as well as the specialty of the algae in plateau lakes. I am also concerned about the reasonability of the antifungal activity because the only antifungal indicator is Fusarium oxysporum.

Minor concerns:

Line 29: Excess “the”.

Line 32: Should be “anti-plant”.

Line 210-216: The result of this part is not displayed, readers could not know where “range of 1–9 μg/ml” are from, also cannot understand why “precision and stability within 2 hours” could come from the corresponding results.

Line 253: There is only one subtitle in the chapter “metabolite identification”, is it necessary to use this subtitle?

Line 286: Need to cite relevant articles or give the exact value of algae and crop yield.

Figure 1: It is recommended to use the colored figure rather than monochrome.

7. PLOS authors have the option to publish the peer review history of their article (what does this mean? ). If published, this will include your full peer review and any attached files.

**Do you want your identity to be public for this peer review?** For information about this choice, including consent withdrawal, please see our Privacy Policy .

Reviewer #3: No

Reviewer #5: No

---

## [Author Response · Author response to Decision Letter 5]

22 Jan 2025

Dear Editor-in-Chief

We would like to have our revised version of manuscript entitled “Extraction, analysis, and antifungal activity study of algae antibiotic active substances in plateau lakes” considered for publication in “Plos one”.

All the comments and suggestions are addressed in the revised version and track

changes are mentioned inside the revised file for editors and reviewers.

Extraction, analysis, and antifungal activity study of algae antibiotic active substances in plateau lakes

Reviewer #3:The manuscript has been improved. The authors could provide the supplimentary data and avail all documents for readers

Author Response: All relevant data have been submitted to the “Plos one” journal. However, no suitable database has been identified for data deposition.

Reviewer #5: The manuscript mainly described how to extract antifungal components from algae and analyzed their antifungal activity. Thus, I think the first thing that the author needs to convince the readers is that the natural antifungal component is better than artificial in some aspects, or the manuscript stands on an unstable base in my point of view. Also, I expect to see more discussions about the advantages and disadvantages of the extraction methods used in this manuscript compared to other methods, as well as the specialty of the algae in plateau lakes. I am also concerned about the reasonability of the antifungal activity because the only antifungal indicator is Fusarium oxysporum.

Author Response: The natural antifungal components derived from algae may not necessarily outperform agents. However, the preparation of phenolic acid-based fungicides using algae from high-altitude lakes addresses two critical concerns: the large-scale utilization of algae resources from these lakes and the management of plant root rot diseases. Therefore, we propose that the preparation of phenolic acid-based fungicides using high-altitude lake algae is economically viable, which has been stated in lines 289-297. In this study, four methods for preparing algal phenolic acid-based fungicides were evaluated. The critical indicators for assessing the advantages and limitations of these methods are the phenolic acid content and the antibacterial activity of the extracted compounds. A detailed analysis of these parameters is presented in Table 1. Fusarium oxysporum is a globally prevalent soil-borne pathogenic fungus with a broad host range. It is known to cause wilt disease in over 100 plant species, including melons, eggplants, bananas, cotton, legumes, and ornamental flowers. Our findings indicate that algal phenolic compounds exhibit significant inhibitory activity against F. oxysporum. These findings highlight their potential for large-scale application as a suitable solution for managing plant fungal diseases.

Minor concerns:

Line 29: Excess “the”.

Author Response: Many thanks for this comment. We have corrected this mistake.

Line 32: Should be “anti-plant”.

Author Response: Thanks, We have corrected it.

Line 210-216: The result of this part is not displayed, readers could not know where “range of 1–9 μg/ml” are from, also cannot understand why “precision and stability within 2 hours” could come from the corresponding results.

Author Response: Thanks for this important suggestion. We have provided a detailed method for determining the phenolic acid content

Line 253: There is only one subtitle in the chapter “metabolite identification”, is it necessary to use this subtitle?

Author Response: Thanks, We have removed the subtitle.

Line 286: Need to cite relevant articles or give the exact value of algae and crop yield.

Author Response: Thanks for this important suggestion. We have provided the cite articles, line 292-295.

Figure 1: It is recommended to use the colored figure rather than monochrome.

Author Response: Many thanks for this comment. Figure 1 has been modified to a color graphic.

---

## [Decision Letter · Decision Letter 5]

11 Feb 2025

Extraction, analysis, and antifungal activity study of algae antibiotic active substances in plateau lakes

PONE-D-24-12637R5

Dear Dr. Jing Zhao,

We’re pleased to inform you that your manuscript has been judged scientifically suitable for publication and will be formally accepted for publication once it meets all outstanding technical requirements.

Kind regards,

Awatif Abid Al-Judaibi, PhD

Academic Editor

PLOS ONE

Additional Editor Comments (optional):

Reviewers' comments:

Reviewer's Responses to Questions

**Comments to the Author**

1. If the authors have adequately addressed your comments raised in a previous round of review and you feel that this manuscript is now acceptable for publication, you may indicate that here to bypass the “Comments to the Author” section, enter your conflict of interest statement in the “Confidential to Editor” section, and submit your "Accept" recommendation.

Reviewer #3: All comments have been addressed

Reviewer #5: All comments have been addressed

2. Is the manuscript technically sound, and do the data support the conclusions?

Reviewer #3: Yes

Reviewer #5: Yes

3. Has the statistical analysis been performed appropriately and rigorously? 

Reviewer #3: N/A

Reviewer #5: Yes

4. Have the authors made all data underlying the findings in their manuscript fully available?

Reviewer #3: Yes

Reviewer #5: Yes

5. Is the manuscript presented in an intelligible fashion and written in standard English?

Reviewer #3: Yes

Reviewer #5: Yes

6. Review Comments to the Author

Reviewer #3: The authors have addressed the comments to the best of their ability. What has not been addressed can fully be incorporated under the recommendations

Reviewer #5: My concerns have been addressed, and I don't see major problems in my area of expertise. However, it is recommended that some of the statements in the manuscript be carefully considered to make them clearer to readers from other areas.

7. PLOS authors have the option to publish the peer review history of their article (what does this mean? ). If published, this will include your full peer review and any attached files.

**Do you want your identity to be public for this peer review?** For information about this choice, including consent withdrawal, please see our Privacy Policy .

Reviewer #3: No

Reviewer #5: No

---

## [Editor Report · Acceptance letter]

PONE-D-24-12637R5

PLOS ONE

Dear Dr. Zhao,

I'm pleased to inform you that your manuscript has been deemed suitable for publication in PLOS ONE. Congratulations! Your manuscript is now being handed over to our production team.

Kind regards,

on behalf of

Professor Awatif Abid Al-Judaibi

Academic Editor

PLOS ONE